# Better Generalization by Efficient Trust Region Method

## Abstract

In this paper, we develop a trust region method for training deep neural networks. At each iteration, trust region method computes the search direction by solving a non-convex subproblem. Solving this subproblem is non-trivial—existing methods have only sub-linear convergence rate. In the first part, we show that a simple modification of gradient descent algorithm can converge to a global minimizer of the subproblem with an asymptotic linear convergence rate. Moreover, our method only requires Hessian-vector products, which can be computed efficiently by back-propagation in neural networks. In the second part, we apply our algorithm to train large-scale convolutional neural networks, such as VGG and MobileNets. Although trust region method is about 3 times slower than SGD in terms of running time, we observe it finds a model that has lower generalization (test) error than SGD, and this difference is even more significant in large batch training. We conduct several interesting experiments to support our conjecture that the trust region method can avoid sharp local minima.

## 1 Introduction

Despite having many optimization algorithms in the literature (Nesterov, 2013; Wright & Nocedal, 1999; Boyd & Vandenberghe, 2004), when talking about neural network optimization, by far the most popular methods are SGD and its variants such as AdaDelta (Zeiler, 2012), AdaGrad (Duchi et al., 2011), Adam (Kingma & Ba, 2014) and RMSProp (Hinton et al., 2006). Compared to these stochastic first order methods, second order methods are rarely used in training deep neural networks due to large data files and high dimensional parameter space. Moreover, neural network has non-convex loss function and its Hessian is often ill-conditioned, making it difficult to apply second order methods. Prior to our work, there aren't many works on second order methods for neural network optimization (Martens, 2010; Kiros, 2013; Botev et al., 2017), and most of them are tested on multi-layer perceptron so it is unclear whether second order methods can be useful for more advanced networks such as convolutional neural networks.

In this paper, we develop an efficient trust region method for training deep neural networks, where the objective function is approximated by a quadratic term with a norm constraint, known as the trust-region subproblem. The main difficulty in large-scale applications is how to solve this *non-convex* subproblem efficiently (Yuan, 2000). Approximate solvers such as (Steihaug, 1983; Powell, 1970) cannot converge to the exact minimizer, while the best method for computing the global minimizer has sub-linear convergence rate (Hazan & Koren, 2016). We propose a new solver that is guaranteed to converge to a global minimum of the nonconvex trust region sub-problem with an asymptotic linear convergence rate. What's more, our method only requires Hessian-vector product, which only costs two forward-backward operations in neural networks. Empirically we find this technique making trust region method comparable to SGD in terms of running time.

Apart from convergence rate, generalization ability is another important issue to practitioners. Starting from (Zhang et al., 2016) which proposed the generalization paradox of deep models, intensive works have been done to find ways to explain and close generalization gap in large batch SGD (Keskar et al., 2016; Kawaguchi et al., 2017; Wilson et al., 2017). Among them, (Keskar et al., 2016) empirically finds the generalization gap between large and small batch and later (Wilson et al., 2017) notices that adaptive methods like Adam, AdaGrad and RMSprop are even worse than SGD concerning large batch generalization. To close the generalization gap, (Chaudhari et al., 2016) changes

---

**Algorithm 1** (Stochastic) Trust region method

---

**Input:** $\{F(x), \nabla F(x), \nabla^2 F(x)\}$, or their stochastic counterparts; Initial guess $x_0$, radius $r_0$.
**Output:** Suboptimal solution: $x_{N+1}$.
1: **for** $t = 0, 1, \cdots, N$ **do**
2:      Calculate gradient $g = \nabla F(x_t)$, $H = \nabla^2 F(x_t)$
3:      Expand objective function:

$$\underset{z:\|z\|_2 \leq r_t}{\arg\min} f(z) = \frac{1}{2} z^\mathsf{T} H z + g^\mathsf{T} z \tag{1}$$

4:      Solve (1) to get $z_t$, compute $q_t = \frac{F(x_t) - F(x_t + z_t)}{f(0) - f(z_t)}$
5:      (Deterministic version) Update iterate and radius:
        • If $q_t \geq 0.9$, then $x_{t+1} = x_t + z_t$, $r_{t+1} = 2r_t$.
        • If $q_t \geq 0.1$, then $x_{t+1} = x_t + z_t$, $r_{t+1} = r_t$.
        • If $q_t < 0.1$, then $x_{t+1} = x_t$, $r_{t+1} = 0.5r_t$.
6:      (Stochastic version) Update iterate and radius:
        • If $q_t \geq 0.9$ and $\|g\| \geq r_t$, then $x_{t+1} = x_t + z_t$, $r_{t+1} = \min\{2r_t, r_{\max}\}$.
        • Otherwise, $x_{t+1} = x_t$, $r_{t+1} = 0.5r_t$.
7: **end for**
8: **return** $x_{N+1}$.

---

the loss function to eliminate sharp local minima, (De et al., 2016) selects batch size adaptively and (Hoffer et al., 2017) combines adaptive step size with new batch normalization layer.

In contrast to these works, we find through experiments that using trust region method alone can effectively escape from sharp minima and achieve lower test error than SGD, especially on large-batch training. For CIFAR-10 with VGG network, our trust region based method achieves 86.8% accuracy and our hybrid method have 88.1%, both of them are better than SGD's solution with 86.7% accuracy when batch size is 128. As batch size grows to 2048, our trust region method still has 85.5% accuracy and our hybrid method is 87.0%, in contrast, SGD only has 80.3% accuracy. Arguably, our method is a more natural solution to large batch training because it can match the small batch accuracy simply by switching to trust region method without changing network structure, loss function and batch size.

## 2 BACKGROUND AND RELATED WORK

### 2.1 (STOCHASTIC) TRUST REGION METHOD

Trust region (TR) method (Conn et al., 2000) (Algorithm 1) is a classical second order method for solving nonlinear optimization problems. At each iteration, TR forms a second order approximation around the current solution and updates the solution by minimizing this quadratic function within the "trust region". The trust region, usually a hyper ball, is chosen to make the second order approximation close enough to the original one and thus by minimizing the subproblem, the objective function will very likely to descent. More formally, suppose $F(x)$ is the objective function, at iteration $x_t$ the subproblem can be written as:

$$\underset{z}{\arg\min} f(z) := \frac{1}{2} z^\mathsf{T} H z + g^\mathsf{T} z, \ \ \text{s.t.} \ \|z\|_2 \leq r. \tag{2}$$

Where $H = \nabla^2 F(x_t)$, $g = \nabla F(x_t)$, and $r > 0$ is the radius of trust region.[1]

If objective function $F$ has finite sum structure, $F(x) = \frac{1}{n} \sum_{i=1}^{n} f_i(x)$ or more generally $F(x) = \mathbb{E}_\xi f(x, \xi)$ where $\xi$ is random variable with bounded variance, then we can turn to stochastic trust region method discussed in (Chen et al., 2015). Both deterministic and stochastic trust region method

---

[1]In general the constraint can be $\|\cdot\|_M$ for any PSD matrix $M$, but we only discuss the simpler case here because a change-of-variable $z' = \sqrt{M}z$ and $H' = \sqrt{M}^{-\mathsf{T}} H \sqrt{M}^{-1}$ can transform it to (2).

have very similar algorithm as one can see in Algorithm 1. As to convergence rate, (Blanchet et al., 2016) shows that the stochastic version is guaranteed to find a stationary point $\|\nabla f(x)\| \leq \epsilon$ in $\mathcal{O}(\epsilon^{-2})$ iterations, while the deterministic version is able to find $\epsilon$-suboptimal stationary point within $\mathcal{O}(\epsilon^{-3/2})$ iterations (Curtis et al., 2017). For large scale problem when we cannot calculate the gradient and Hessian on the whole dataset, stochastic method is the only choice.

To apply trust region method to large-scale non-convex optimization (e.g., training deep networks), there are two difficulties: (a) The Hessian matrix $H$ may not be positive definite, so (2) is a non-convex problem with constraint and it is hard to compute its global minimizer. (b) $H$ is very large (millions by millions) and cannot be formed explicitly. For (a), several algorithms have been proposed in the literature. Traditionally, dogleg method (Powell, 1970) for $H \succ 0$ and Steihaugs method (Steihaug, 1983) for indefinite $H$ can be applied to get an approximated solution. However if a more precise solution is desired, we need to find other ways, e.g. (Hazan & Koren, 2016) is able to find an $\epsilon$-suboptimal solution in $\tilde{\mathcal{O}}(M/\sqrt{\epsilon})$ time, where $M$ is the matrix size (number of nonzero elements). In this paper, we design an efficient solver based purely on gradient information that can find an $\epsilon$-suboptimal solution in asymptotically linear time $\tilde{\mathcal{O}}(M \log(1/\epsilon))$.

For (b), due to high dimensionality of $H$, matrix decomposition such as SVD will be in-feasible. Therefore, an efficient subproblem solver should only involve computations of Hessian-vector product, which can be computed by back-propagation in deep networks (Pearlmutter, 1994; Baydin et al., 2015). Furthermore, since machine learning problems can usually be formulated as finite sum structure $F(x) = \frac{1}{N} \sum_{i=1}^{N} f_i(x)$, instead of computing the exact Hessian we can compute the subsampled Hessian on the selected batch, so the subsampled Hessian-vector product computation is only slower than SGD computation by a constant factor. Our proposed subproblem solver only involves Hessian-vector product, so it can be applied to large-scale problems and replace SGD seamlessly. This subsampled version is justified by the stochastic trust region method described above.

## 2.2 Other Second Order Methods for Training Neural Networks

Another related technique for non-convex optimization is adaptive regularization with cubic (ARC) (Griewank, 1981). Similar to TR method, ARC uses cubic term of positive coefficient $\frac{\sigma}{3} \|z\|_2^3$ to replace constraint in (2):

$$\arg\min_z f(z) := \frac{1}{2} z^\mathsf{T} H z + g^\mathsf{T} z + \frac{\sigma}{3} \|z\|_2^3. \tag{3}$$

ARC method can be seen as the "twin" method of TR, since they share many similar properties (e.g. converge rates). Intuitively the radius parameter $r$ in (2) plays the same role as $\sigma$ in ARC: a larger $r$ and smaller $\sigma$ both mean larger "trust-region". However, since the ARC subproblem is unconstrained, the treatment is relatively easy: for example, generalized Lanczos-type iterations (Gould et al., 1999) or gradient descent (Carmon & Duchi, 2016) can be used. Our proposed subproblem solver for TR is also based on gradient descent, but due to constraint in the TR subproblem our algorithm and proof are different from (Carmon & Duchi, 2016).

Recently, several works have applied second order methods to neural network optimization, including (Martens, 2010; Botev et al., 2017; Xu et al., 2017b;a). However, all of them only consider small-scale fully connected networks with few layers. Among them, (Xu et al., 2017b;a) adopt full gradient and subsampled Hessian and solve the trust region subproblem in the subspace spanned by Cauchy point and minimum eigenvector $v_1(H)$. We are different from this recent paper in two aspects. First, we develop a new trust region subproblem solver with better theoretical guarantee. Second, the full gradient computation in their algorithm is too expensive for large-scale deep network training. In contrast, we implement our algorithm with subsampled gradient and Hessian, thus we are able to run large-scale experiments and observe better generalization compared with SGD.

## 2.3 Generalization Ability for First Order Methods

SGD and other stochastic first order methods are popular for training large scale neural networks since gradient for each training sample can be computed in one forward-backward cycle. Recently there are many works on the effect of batch size on training time as well as generalization ability. Indeed, a larger batch makes more use of parallel processors, so the program can scale perfectly to

more GPUs. Moreover, more samples make the gradient less noisy and thus a larger learning rate and higher converge rate are possible, see (Goyal et al., 2017; You et al., 2017a;b) and references therein.

At the same time, large batch in SGD is harmful to generalization ability, experiments in (Keskar et al., 2016) show that batch size can affect the geometry around the solution that SGD finds. Specifically, SGD with a larger batch size tends to find sharper local minima, so even a tiny change of data from training set to test set will lead to high loss, i.e. bad generalization ability. As to the converge rate to local minima, although (Lee et al., 2016) shows gradient descent always converges to local minima, (Du et al., 2017) proves the converge process can take exponential time. On the other hand by adding extra noise, (Ge et al., 2015) claims SGD is able to escape strict saddle in $\mathrm{poly}(d/\epsilon)$ iterations, and further (Jin et al., 2017) design a full gradient with noise method that improves the result to $\mathrm{poly}\text{-}\log(d)/\epsilon^2$ steps.

**Notations**: Throughout this paper, we use $\mathcal{S}^{n-1}$ to denote the unit sphere in $\mathbb{R}^n$. The model parameters $x \in \mathbb{R}^d$ and objective function $F(x) = \frac{1}{N}\sum_{i=1}^{N} f_i(x)$ is in $\mathcal{C}^2$; its gradient and Hessian at step $x_t$ are denoted as $g_t = \nabla F(x_t)$ and $H_t = \nabla^2 F(x_t)$, and if the index $t$ is unimportant we will simplify them as $g$ and $H$ respectively. For the Hessian matrix $H$, its eigenvalue decomposition is $H = \sum_{i=1}^{d} \lambda_i v_i v_i^\mathsf{T}$ such that $v_i^\mathsf{T} v_j = \delta_{ij}$ and $\lambda_1 \le \lambda_2 \le \cdots \le \lambda_d$, accordingly we decompose any vector $a \in \mathbb{R}^d$ as $a = \sum_{i=1}^{d} a^{(i)} v_i$. Denote operator norm $\beta = \|H\|_{\mathrm{op}} = \max\{|\lambda_1|, |\lambda_d|\}$. $\|\cdot\|$ represents 2-norm if not stated explicitly, $\mathrm{I}_d \in \mathbb{R}^{d \times d}$ is the identity matrix. Finally, following (Carmon & Duchi, 2016) we use $s$ to represent the global minimizer of trust region subproblem (2).

## 3 EFFICIENT TRUST REGION SUBPROBLEM SOLVER

---

**Algorithm 2** Proposed trust region subproblem solver

---
**Input:** Gradient $g$, Hessian matrix $H$
**Output:** Approximated solution $z$
 1: Initialize $z_0$ and $\eta$ so that Assumption 1,2 are met.
 2: `boundary=false`
 3: **for** $t = 0, 1, \ldots, N_1$ **do**
 4:     **if** $\|z_t\| < 1$ **then**
 5:         $z_{t+1} = z_t - \eta(Hz_t + g)$
 6:     **else**
 7:         `boundary = true`
 8:         **break**
 9:     **end if**
10: **end for**
11: **if** `boundary = false` **then return** $z_{N_1+1}$
12: **end if**
13: **for** $t' = t, t+1, \ldots, N_2$ **do**
14:     Choose $\alpha_{t'}$ by Armijo line search
15:     $z_{t'+1} = \dfrac{z_{t'} - \alpha_{t'}(\mathrm{I}_d - z_{t'} z_{t'}^\mathsf{T})(Hz_{t'}+g)}{\|z_{t'} - \alpha_{t'}(\mathrm{I}_d - z_{t'} z_{t'}^\mathsf{T})(Hz_{t'}+g)\|}$
16: **end for**
17: **return** $z_{N_2+1}$

---

In this section, we show an efficient gradient-descent based algorithm with proper initialization can find the global minimum of the trust region subproblem (2). If the global minimum lies inside of the sphere then gradient descent itself is guaranteed to find it; Otherwise we first conduct gradient descent until the iterate hits the spherical constraint, then a manifold gradient descent on the sphere can converge to the solution. We prove this simple procedure can return the global minimum of the non-convex trust region subproblem and has asymptotically linear convergence rate. The details are shown in Algorithm 2.

### 3.1 PROPERTIES OF GLOBAL MINIMUM

The necessary and sufficient condition of global minimum comes from KKT condition, see (Wright & Nocedal, 1999) for details.

**Lemma 1.** *(Global minimum) $s$ is the global minimum of* (2) *if and only if $\|s\| \leq 1$ and there is a scalar $\lambda \geq 0$ such that:*

$$
\begin{aligned}
\text{gradient condition: } & (H + \lambda I_d)s + g = 0, \\
\text{complementary slackness: } & \lambda(1 - \|s\|_2) = 0, \\
\text{Hessian: } & H + \lambda I_d \succeq 0.
\end{aligned}
\tag{4}
$$

**Proposition 1.** *Based on* (4) *we can describe the solution(s) as follows:*

- $\lambda_1 > 0$: *only one global minimum. If $\|H^{-1}g\| < 1$ then $\|s\| < 1$, otherwise $\|s\| = 1$.*

- $\lambda_1 = 0$: *if $g^{(1)} \neq 0$, only one solution with $\|s\| = 1$; Otherwise if $g^{(1)} = 0$ and $\sum_{i=2}^{n}(\frac{g^{(i)}}{\lambda_i})^2 \geq 1$, one solution with $\|s\| = 1$; If $g^{(1)} = 0$ and $\sum_{i=2}^{n}(\frac{g^{(i)}}{\lambda_i})^2 < 1$, there are multiple solutions.*

- $\lambda_1 < 0$, $g^{(1)} \neq 0$: *only one solution.*

- $\lambda_1 < 0$, $g^{(1)} = 0$: *if $\sum_{i=2}^{n}(\frac{g^{(i)}}{\lambda_i - \lambda_1})^2 \geq 1$ then only one global minimum, otherwise there are multiple ones.*

Proof details are postponed to appendix. Since when $\lambda_1 \geq 0$ every stationary points are global minimum, and further if $\|s\| < 1$ then we only need to use gradient descent to solve it (that the case for Line 11 in Algorithm 2), otherwise $\|s\| = 1$ and our manifold based algorithm also applies, since in this case subproblem is strongly convex, we don't need to worry about converging to suboptimal point. Now we only need to consider about $\lambda_1 < 0$ case as follows. Notice that in this case we always have $\|s\| = 1$. In the following lemma we try to distinguish global minimum from other stationary points, restricted to $g^{(1)} \neq 0$ case (we call it "easy-case", as oppose to "hard-case" in (Wright & Nocedal, 1999)). As to the "hard-case" $g^{(1)} = 0$, in theory we can apply a small perturbation to $b$ as (Carmon & Duchi, 2016) does: $b' = b + \varepsilon$ where $\varepsilon$ is a small Gaussian noise. In practice due to rounding error this case is hardly seen, for best efficiency we choose to ignore it.

The following lemma gives a sufficient condition of global minimum:

**Lemma 2.** *When $g^{(1)} \neq 0$ and $\lambda_1 < 0$, among all stationary points if $s^{(1)}g^{(1)} \leq 0$ then $s$ is the global minimum.*

### 3.2 CONVERGENCE OF ITERATION

Now it left to see how the solution found by Algorithm 2 will meet Lemma 2. To this end, we need to enforce following assumptions:

**Assumption 1.** *(Bounded step size) Step size $\eta < 1/\beta$.*

**Assumption 2.** *(Initialize) $z_0 = -\alpha \frac{g}{\|g\|}$, $0 < \alpha < 1$.*

We remark that Assumption 2 is a good guess of global minimum if ignoring the curvature information, and using this alone as subproblem solution will reduce trust region method to gradient descent. Under these assumptions, Algorithm 2 is guaranteed to find (one of) the global minimum, which is formally stated in the following theorem:

**Theorem 3.** *Under proximal gradient descent update: $z_{t+1} = \mathsf{Prox}_{\|\cdot\|}(z_t - \eta \nabla f(z_t))$, and Assumption 1 if $z_t^{(i)}g^{(i)} \leq 0$ then $z_{t+1}^{(i)}g^{(i)} \leq 0$. Combining with Assumption 2 and Lemma 2, if $\lambda_1 < 0$, $g^{(1)} \neq 0$ then $z_t$ converges to a global minimum $s$.*

To see the converging process more clearly, we divide the iterations into two phases: in the first phase $z_t$ stays strictly inside the sphere $\mathcal{S}^{n-1}$: $\|z_t\| < 1$, and during this phase we will show that $\|z_t\|$ is monotone increasing until it hits the sphere and that is when the second phase begins. In the second phase the iterates adhere to the sphere and converge to the global minimum with asymptotically linear rate. First of all we show the monotone increasing property based on following lemma:

**Lemma 4.** *For $z_t$ defined above, we have $z_t^\mathsf{T} H \nabla f(z_t) \geq \beta z_t^\mathsf{T} \nabla f(z_t)$ (recall we define $\beta$ as the operator norm of $H$).*

Now *imagine* there is another iterate $\tilde{z}_t$ that does "plain" GD (i.e. without projection): $\tilde{z}_{t+1} = \tilde{z}_t - \eta \nabla f(\tilde{z}_t)$, using the same step size as $z_t$ and same initialization $\tilde{z}_0 = z_0$. Such a iteration rule guarantees that as long as $\|z_t\| < 1$ for all $z \leq \tau$ then $\tilde{z}_t = z_t$. By showing $\|\tilde{z}_t\|$ is monotone increasing, we actually prove monotone increasing property in the first phase.

**Theorem 5.** *Suppose $z_t$ is in the region such that proximal gradient update equals to plain GD: $\tilde{z}_{t+1} = \tilde{z}_t - \eta \nabla f(\tilde{z}_t)$, then under this update rule, $\|\tilde{z}_t\|$ is monotone increasing.*

Another key observation from Theorem 5 is the unboundedness of $\|\tilde{z}_t\|$, i.e. $\|\tilde{z}_t\| \to \infty$ as $t \to \infty$. Indeed we have the following lemma:

**Lemma 6.** *(Finite phase I) Assuming $\lambda_1 < 0$, suppose $t^*$ is the index that $\|\tilde{z}_{t^*}\| < 1$ and $\|\tilde{z}_{t^*+1}\| \geq 1$, then $t^*$ is bounded by:*

$$t^* \leq \log(1 - \eta\lambda_1)^{-1} \Big[ \log\big(\frac{1}{\eta|g^{(1)}|} - \frac{1}{\eta\lambda_1}\big) - \log\big(\frac{-\tilde{z}_0^{(1)}}{\eta g^{(1)}} - \frac{1}{\eta\lambda_1}\big) \Big]. \tag{5}$$

So after $t^*$ iterations the algorithm will reach phase II. Then we turn to manifold optimization method, which is discussed extensively in (Absil et al., 2009) and for the manifold theory we refer (Do Carmo & Flaherty Francis, 1992). Here we list some concepts and its explicit form in our problem: denote $\mathcal{M}$ as the smooth manifold, and $z \in \mathcal{M}$ can be any point on manifold $\mathcal{M}$. The *tangent space* of $z$, denoted as $\mathcal{T}_z$, is the set of all tangent vectors to $\mathcal{M}$ at $z$. When $\mathcal{M} = \mathcal{S}^{n-1}$ then $\mathcal{T}_z = \{\xi \in \mathbb{R}^n : z^\mathsf{T}\xi = 0\}$; *Retraction* is a mapping $R_z(\xi) : \xi \in \mathcal{T}_z \mapsto \mathcal{M}$, in $\mathcal{S}^{n-1}$ manifold, one of the commonly used retraction is $R_z(\xi) = \frac{z+\xi}{\|z+\xi\|}$; *Projection* onto $\mathcal{T}_z$, denoted as $P_z\xi$ is a mapping from $\mathbb{R}^n$ to $\mathcal{T}_z$, for $\mathcal{S}^{n-1}$ the projection is simply $P_z(\xi) = (\mathrm{I}_d - zz^\mathsf{T})\xi$. Based on above concepts, we can write the gradient descent on $\mathcal{M}$ as: $z_{t+1} = R_{z_t}(-\alpha_t \mathrm{grad} f(z_t))$, and $\alpha_t$ is tactically selected by line search such that $f(z_t) - f(z_{t+1}) \geq \sigma\alpha_t \|\mathrm{grad} f(x_t)\|$, where $\sigma \in (0,1)$.

Because we already know at least one of the global minimum lies on $\mathcal{S}^{n-1}$, and according to Lemma 6 after at most $t^*$ iterations we start to use manifold based optimization, i.e. the subproblem becomes:

$$\min_{z \in \mathcal{S}^{n-1}} f(z) = \frac{1}{2}z^\mathsf{T} Hz + g^\mathsf{T} z. \tag{6}$$

Therefore, our method shrinks the search space from $\|z\| \leq 1$ to $\|z\| = 1$, by doing so we can apply well studied manifold optimization theory (Absil et al., 2009; Udriste, 1994) to our problem. Indeed we have following theorems:

**Theorem 7.** *Let $\{z_t\}$ be an infinite sequence of iterates generated by line search gradient descent, then every accumulation point of $\{z_t\}$ is a stationary point of the cost function $f$.*

This above only guarantees convergence to stationary points, however, according to Theorem 3 if the step size $\alpha_t$ is not too large, it actually converges to global minimum. Then it remains to show a linear convergence rate, as guaranteed by the following theorem:

**Theorem 8.** *Let $\{z_t\}$ be an infinite sequence of iterates generated by line search gradient descent, suppose it converges to $s$. Let $\lambda_{H,\min}$ and $\lambda_{H,\max}$ be the smallest and largest eigenvalues of the Hessian at $s$. Assume that $s$ is a local minimizer then $\lambda_{H,\min} > 0$ and given $r$ in the interval $(r_*, 1)$ with $r_* = 1 - \min\big(2\sigma\bar{\alpha}\lambda_{H,\min}, 4\sigma(1-\sigma)\beta\frac{\lambda_{H,\min}}{\lambda_{H,\max}}\big)$, there exists an integer $K$ such that:*

$$f(z_{t+1}) - f(s) \leq r\big(f(z_t) - f(s)\big),$$

*for all $t \geq K$.*

**Remarks**: $\lambda_{H,\min}$ and $\lambda_{H,\max}$ are the minimum and maximum eigenvalue of Riemannian Hessian. Specifically the Riemannian Hessian can be calculated by(Proposition 5.5.4, Absil et al. (2009)):

$$\begin{aligned}
\mathrm{Hess}\, f(x) &= \mathrm{Hess}(f \circ \mathrm{Exp}_x)(0_x), \\
\langle \mathrm{Hess}\, f(x)[\xi], \xi\rangle &= \langle \mathrm{Hess}\,(f \circ \mathrm{Exp}_x)(0_x)[\xi], \xi\rangle.
\end{aligned} \tag{7}$$

By direct calculation(in appendix) we shall see: $\langle \text{Hess} f(s)[\xi], \xi \rangle = -s^\intercal H s + \xi^\intercal H \xi - g^\intercal s$, where $\xi \in \text{Null}(s), \|\xi\| = 1$. By optimal condition (4), we have:

$$s^\intercal (H + \lambda I_d) s + s^\intercal g = 0 \Rightarrow -s^\intercal H s - g^\intercal s = \lambda, \tag{8}$$

so $\langle \text{Hess} f(s)[\xi], \xi \rangle = \lambda + \xi^\intercal H \xi \geq \lambda + \lambda_1 \overset{!}{>} 0$. Where $\overset{!}{>}$ is guaranteed by gradient condition in (4):

$$(\lambda_1 + \lambda) s^{(1)} + g^{(1)} = 0, \tag{9}$$

in "easy-case", $g^{(1)} \neq 0$, so $\lambda_1 + \lambda \neq 0$ and Hessian condition in (4) can be improved to $\lambda_1 + \lambda > 0$.

Based on above discussion, we know $\lambda_{H,\min} \geq \lambda + \lambda_1 > 0$ and $\lambda_{H,\max} \leq \lambda + \lambda_n$.

## 4 EXPERIMENTS

In this section, we examine the performance of our algorithm. First of all, we use a random generated problem to check the dynamics of our proposed trust region subproblem solver, then we compare the performance of our trust region method with SGD on deep convolutional neural network. After that, we focus on generalization ability of the solution that trust region method returns. We include following algorithms in our experiments:

- **SGD**: We tune the step size to get the fastest, stable convergence.
- **TR**: Trust region method with our proposed subproblem solver. For each subproblem, we choose $N_1 = N_2 = 1$ (defined in Algorithm 2). The trust region radius is set to 1 and the step size of subproblem solver is set to 0.1.
- **Hybrid**: Our proposed hybrid method, it runs trust region method until the accuracy no longer increases, and then uses the last epoch to initialize SGD method. When the training accuracy stabilizes, it returns the final model. Both TR and SGD here follow the same hyper-parameters above.

### 4.1 SOLVING THE TRUST REGION SUBPROBLEM

We sample an indefinite random matrix by $H = BB^\intercal - \lambda I_n$, where $B \in \mathbb{R}^{n \times (n-1)}$ and $B_{ij} \overset{\text{iid}}{\sim} \mathcal{N}(0,1)$, obviously $\lambda_{\min}(H) = \lambda_1 = -\lambda$. Afterwards we sample a vector $g$ by $g_i \overset{\text{iid}}{\sim} \mathcal{N}(0,1)$. By changing the value of $\lambda$ in $\{10, 30, 50, 70, 90, 110\}$, we plot the function value decrement with respect to number of iterations in Figure 1(left). As we can see, the iterates first stay inside of the sphere (phase I) for a few iterations and then stay on the boundary (phase II). To inspect how $\lambda$ changes the duration of phase I, we then plot the number of iterations it takes to reach phase II, under different $\lambda$ values shown in Figure 1(middle). Recall in (5), number of iterations is bounded as a function of $\lambda$, which can be further simplified to:

$$t^* \leq \frac{\log(1 + \frac{\lambda}{|g^{(1)}|})}{\log(1 + \eta \lambda)} = \frac{\log(1 + c_1 \lambda)}{\log(1 + c_2 \lambda)}, \tag{10}$$

where we set $\tilde{z}_0^{(1)} = 0$ to simplify the formula. By fitting the data point with function $T(\lambda) = \frac{\log(1 + c_1 \lambda)}{\log(1 + c_2 \lambda)}$, we find our bounds given by Lemma 6 is quite accurate.

### 4.2 OPTIMIZING DEEP CNNS

Next, we test the performance of our proposed trust region method (actually stochastic is used, denoted as TR) on training deep CNNs, including VGG (Simonyan & Zisserman, 2014) and MobileNets (Howard et al., 2017). VGG/MobileNets are adopted to classify CIFAR10/STL10 data. During all experiments, we only compare trust region method with plain SGD. We did not include other SGD variants because (1) We want to focus on whether second order information is useful without considering the effect of momentum/acceleration. (2) It is known that SGD generalize better than adaptive methods in large-batch (Wilson et al., 2017). For reference, we also list the best accuracy obtained by Adam with small batch size in Table 1, which shows Adam's accuracy is only slightly better than SGD but still worse than our hybrid method.

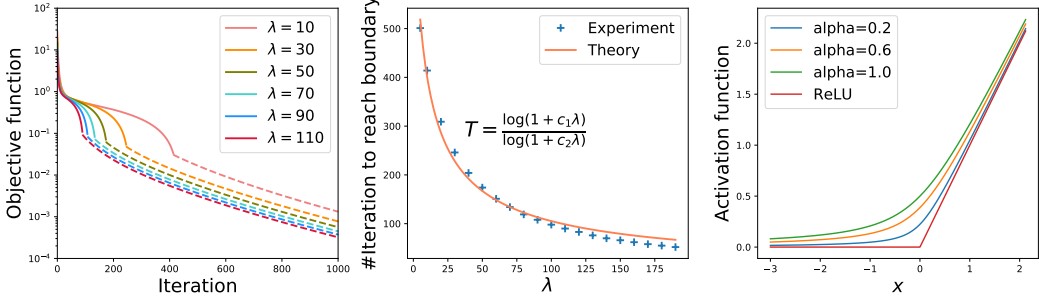

Figure 1: *Left*: Trust region experiment, we use solid lines to indicate iterations inside the sphere and dash lines to indicate iterations on the sphere. By changing $\lambda$ we can modify the function curvature. *Middle*: #Iteration it takes to reach sphere under different $\lambda$'s, we also fit the curve by model $T = \frac{\log(1+c_1\lambda)}{\log(1+c_2\lambda)}$ derived in Lemma 6. *Right*: Comparing ReLU with SReLU (see definition in Section 4.2), as $\alpha$ closes to zero, SReLU looks more like ReLU.

We have some tricks to make it possible to apply trust region method to a deep convolutional network: First of all, rather than taking ReLU as the activation function, we use a similar yet everywhere continuously differentiable function, namely SReLU (see Figure 1(right)):

$$\text{SReLU}(x) = \frac{\sqrt{x^2 + \alpha} + x}{2}, \quad \alpha > 0.$$

Although we can also define Hessian on ReLU function, it is not well supported on major platforms (Theano/PyTorch). Likewise, we find max-pooling is also not supported by platforms to calculate higher order derivative, one way to walk around is to change all the max-pooling layers to avg-pooling—it hurts accuracy a little bit (we observe 1~2% accuracy loss compared with (Keskar et al., 2016)), albeit this is not our primary concern. Secondly, the oracle in TR method is Hessian-vector product, or $Hv$. Theoretically, the cost of $Hv$ is comparable to gradient oracle because $Hv = \frac{\partial}{\partial w}(v^\intercal \frac{\partial f}{\partial w})$. However, due to inefficient implementation, in many deep learning frameworks the cost of $Hv$ is ~7x slower than gradient oracle, making running time comparison unfavorable, so we use the following numerical differentiation to replace chain-rule:

$$\widehat{Hv} = \big(\nabla f(x + \varepsilon v) - \nabla f(x)\big)/\varepsilon, \text{ where } \varepsilon \to 0.$$

Experiments show the relative error is controllable ($\|\widehat{Hv} - Hv\|/\|Hv\| \leq 1\%$). Nevertheless, the Hessian-vector operation is still a bottleneck to performance, thus if we care about running time then 1~2 inner iteration(s) for each subproblem is more suitable. Lastly, although in theory we need full gradient and full Hessian to guarantee convergence (in Xu et al. (2017b) they only need full gradient), calculating them in each iteration is not practical, so we calculate both Hessian and gradient on subsampled data to replace the whole dataset.

In the first experiment, we compare the training/testing accuracy w.r.t. running time and epoch on VGG16+CIFAR10/STL10 dataset. For SGD we choose step size $\eta = 0.1$, which is the best step size in $\{10^{-n} \mid n = 1, 2, \dots\}$. For trust region subproblem we choose step size $\eta' = 0.1$ for all the experiments. Both of them use batch size $B = 512$(for CIFAR10) and $B = 1024$(for STL10). Our machine has 4 Titan Xp GPUs and Xeon E5-2620 CPU. The experimental results are shown in Figure 2. Without surprising, the trust region method is ~3x slower than SGD, partly because we use subsampled gradient/Hessian (currently there is no theoretical result to guarantee convergence in this situation) and $Hv$ operation is too expensive. Despite the slower training time, we observe TR converges to a solution with better test error.

### 4.3 BETTER GENERALIZATION ABILITY IN LARGE BATCH TRAINING

Training neural network with larger batch size has become an important issue for faster training. To fully exploit the computational power of multi-GPU systems, we need to increase the batch size,

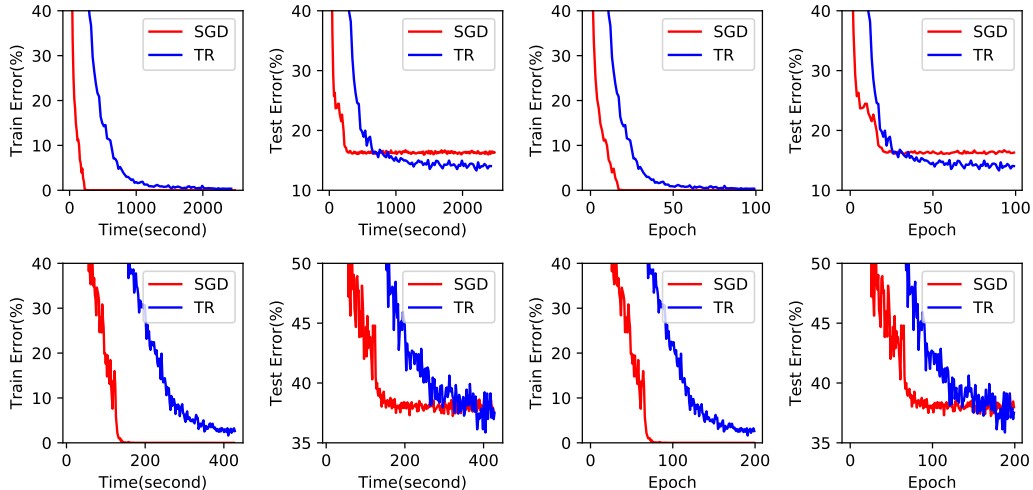

Figure 2: Training/Testing error with both time and epoch. The first row is CIFAR10 dataset(batch size $B = 512$) and the second row is STL10 dataset(batch size $B = 1024$). We find TR method has better test accuracy on both datasets, although the running time is 2∼3x longer.

at least proportional to number of cores. However, large batch size can affect the generalization ability (Keskar et al., 2016), so it would be interesting to see if this dilemma can be solved merely by choosing a different optimization method.

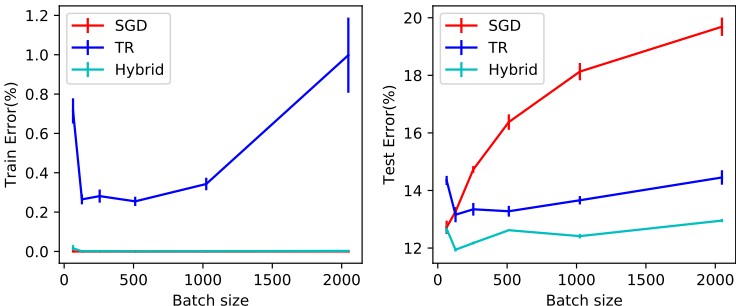

Figure 3: We choose batch size from $\{64, 128, 256, 512, 1024, 2048\}$, for each batch size and each algorithm, we independently run 5 times to gather mean accuracy and deviation (in order to show significance). The experiment is conducted on VGG16+CIFAR10.

To do so, we design an experiment that compares testing accuracy under different batch sizes in Figure 3. We observe that our method (TR) converges to solutions with much better test error when batch size is larger than 128. We postulate this is because SGD has not enough noise to escape from a sharp local minimum, especially when the large batch is used. For smaller batch size, TR method is not as good as SGD method, this is because the approximation of gradient and Hessian is not good enough by only a few samples. However, our Hybrid method combines the benefits of TR and SGD methods to make it both fast and accurate, for both small and large batch size.

### 4.4 LOSS LANDSCAPE OF DIFFERENT SOLUTIONS

To explore whether TR/Hybrid methods find the wide local minimum, we follow the experiment techniques in (Keskar et al., 2016) to draw the loss curve that connects two models. Specifically, if we use $w^\star_{\text{SGD}}$, $w^\star_{\text{TR}}$ and $w^\star_{\text{Hybrid}}$ to denote models of SGD, trust region and hybrid method respectively,

then the interpolated model in between is

$$w^\star = \alpha w_A^\star + (1-\alpha)w_B^\star, \quad \alpha \in [0,1], \quad A, B \in \{\text{SGD}, \text{TR}, \text{Hybrid}\}. \tag{11}$$

The loss and accuracy can be evaluated on both training set and test set by parameters $w^\star$, and the result is displayed in Figure 4. We also calculate the $\ell_2$-distance between model A and B, shown in titles of sub-figures.

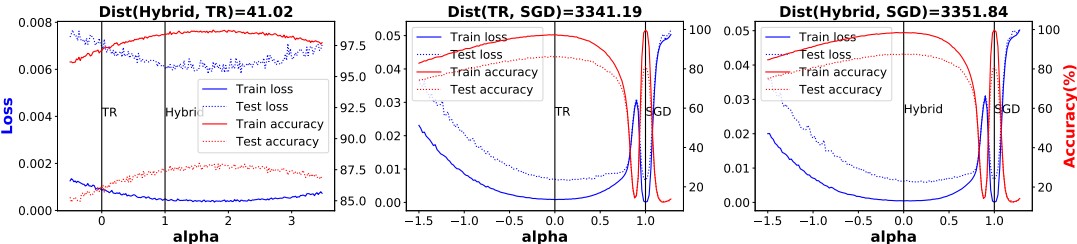

Figure 4: Loss(blue lines) and accuracy(red lines) curve of interpolated models, the x-label is $\alpha$ defined in (11), y-label in the left is the averaged loss and y-label in the right is accuracy. We also calculate the distance of models between $\alpha = 0$ and $\alpha = 1$ in the titles.

Figure 4 clearly shows the width and relative location of local minima: models of TR and hybrid methods lie in the same wide local minimum, while SGD finds a much sharper local minimum. Moreover, we notice Dist(TR, SGD) $\approx$ Dist(Hybrid, SGD) $\gg$ Dist(Hybrid, TR), implying that hybrid method is a refinement of TR method (both of them are close to the exact wide local minimum, but hybrid method is even closer). In contrast, SGD converges to a very different solution, where the loss curve is sharp and the model does not generalize as good as the other two.

### 4.5 Properties of Large Batch SGD vs TR

By far, we can only conclude that trust region method generalize better than SGD, especially when batch size is large. In what follows, we design a novel experiment that help us identifying the differences between those two methods. Rather than running a homogeneous algorithm, we interleave SGD and TR as two tracks, for simplicity we denote SGD($n$) as running $n$ epochs SGD, and TR($n$) as running $n$ epochs TR. In the first track, we run SGD(50) in the beginning, then we use the last iteration in SGD(50) to initialize TR(100), lastly based on the latest iterate, run SGD(50) again. In the second track we change the order: Begin with TR(100), then SGD(50), and end with TR(50).

The purpose of switching between TR and SGD methods is to detect the character of respective solutions they find. Before running experiment we do such "thought experiment": **Imagine** if large batch SGD converges to a sharp local minimum, and TR method can successfully escape it, then initialized by the solution that SGD finds, TR method will "climb over" the loss hill and down to a wide minimum, during this process the loss function (as well as accuracy) first increases and then decreases. On the contrary if TR method already finds a wide local minimum, then initialized by that solution SGD shall not escape to other minima, so we don't expect a sharp change in either loss or accuracy. This thought experiment is illustrated in Figure 5.

Our experimental results are presented in Table 1, where we run VGG16+CIFAR10 and choose batch size $B = 2048$. The overall running process is shown in the picture, and we further extract some important data into the table in order to see the differences more clearly.

We explain the results from following angles:

1. As expectation, there is a sharp raise in both train loss and test error at the 50th epoch in SGD(50)-TR(100)-SGD(50) track, meaning trust region method does escape from the solution that SGD(50) converges to. Meanwhile both loss function and test error change smoothly at the 100th epoch in TR(100)-SGD(50)-TR(50) track, which is understandable because SGD can't escape the basin of wide minimum obtained by TR.

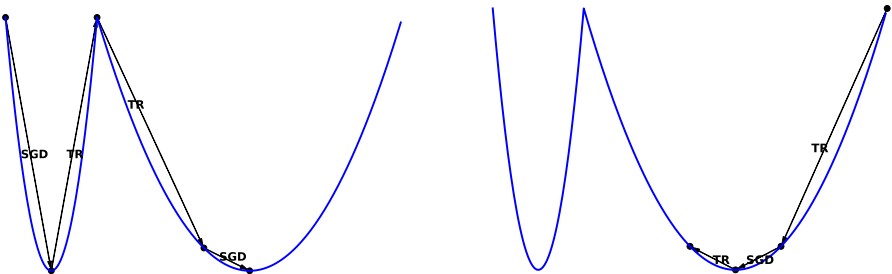

Figure 5: Illustration of **imaginary process** of Track 1(left) and Track 2(right). Note that we use subsampled Hessian and gradient, so the iterate of trust region method will fluctuate around local minimum.

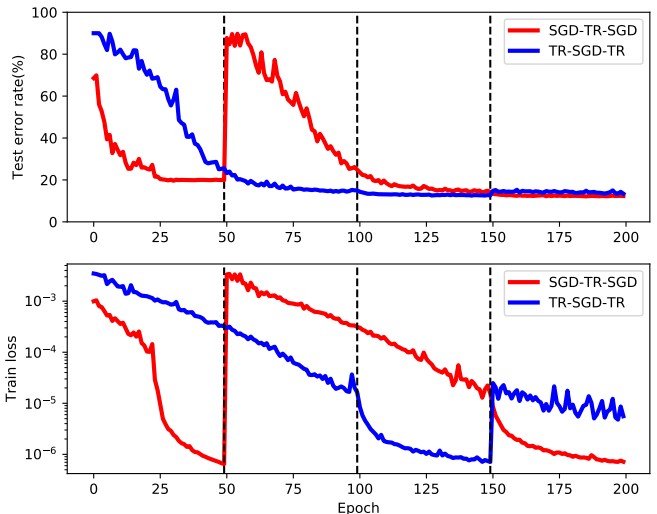

| | Stage I | | Stage II | | Stage III | |
|---|---|---|---|---|---|---|
| | method | accuracy | method | accuracy | method | accuracy |
| Track 1 | SGD(50) | 80.1% | TR(100) | 86.2% | SGD(50) | 87.8% |
| Track 2 | TR(100) | 85.6% | SGD(50) | 87.5% | TR(50) | 86.5% |
| Adam | ADAM(100) | 86.6% | | | | |

Table 1: Testing accuracy in different stages. In Track 1 we run SGD(50)→TR(100)→SGD(50), in Track 2 we change the order to TR(100)→SGD(50)→TR(50), so both tracks run 200 epochs in total. Testing accuracy are reported by the end of each stage. For both tracks we choose batch size $B = 2048$ and CIFAR-10 dataset. As a reference, we also run Adam algorithm on small batch ($B = 64$).

2. Warm-started by TR(100), SGD can reach the best testing accuracy, moreover, the accuracy merely change by enlarging batch size. This sets the foundation of our hybrid method in Figure 3, where the testing accuracy only drops $0.35\%$ from $B = 64$ to $B = 1024$, in comparison SGD testing accuracy drops $6.97\%$.

3. Due to subsampled Hessian and gradient, TR method can only fluctuate around the minimizer, this causes a $1.6\%$ drop in testing accuracy compared to the best result. Another clue is the $1.0\%$ drop in testing accuracy and raise of training loss at 150th epoch in TR(100)-SGD(50)-TR(50) track, when SGD(50)→TR(50) happens.

We want to emphasize that even though we fixed the step size in SGD while in practice we should damp the step size after every few epochs, this won't change the results above, since a smaller step size makes it even harder to escape from sharp minima. In appendix we have supplementary experiments by using other networks combined with other datasets to show that these findings are not specific to one network.

## 5 DISCUSSION

In this paper we first show that a simple gradient based method can effectively find the global minimum of trust region subproblem, even if it is nonconvex. By examining the convergence rate as well as generalization ability, we find our algorithm is comparable to SGD with respect to running time, but can converge to a solution with better generalization error. We suggest to combine trust region with SGD to enjoy both fast and accurate properties. As an important future direction, it would be interesting to see why trust region based algorithm can escape sharp local minima and try to establish convergence results on subsampled trust region method.

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

## A  PROOFS

### A.1  PROOF OF PROPOSITION 1

*Proof.* For clarity, we repeat (4) in Lemma 1 here:

$$
\begin{aligned}
(H + \lambda \mathrm{I}_d)s + g &= 0, \\
\lambda(1 - \|s\|_2) &= 0, \\
H + \lambda \mathrm{I}_d &\succeq 0.
\end{aligned}
\tag{12}
$$

i) If $\lambda_1 > 0$ then we always have $s = -(H + \lambda \mathrm{I}_d)^{-1}g$, since $\|s\| \le 1$, if $\|H^{-1}g\| > 1$ then it must be that $\lambda > 0$ and $\|s\| = \|(H + \lambda \mathrm{I}_d)^{-1}g\| = 1$.

ii) If $\lambda_1 = 0$ and $g^{(1)} \ne 0$, considering about $\lambda s^{(1)} + g^{(1)} = 0$ we can infer $\lambda \ne 0$ and further $\|s\| = 1$. So in this case the only solution is $s = -(H + \lambda \mathrm{I}_d)^{-1}g$ where $\lambda$ is the solution of $\|(H + \lambda \mathrm{I}_d)^{-1}g\| = 1$.

iii) If $\lambda_1 = 0$ and $g^{(1)} = 0$, in this case, either $\lambda = 0$ or $s^{(1)} = 0$. If $\lambda = 0$ then $Hs + g = 0$, this is equivalent to $s^{(i)} = -\frac{g^{(i)}}{\lambda_i}$ for $i \ge 2$ (suppose $\lambda_i > 0$ since $i \ge 2$). On the other hand $\|s\| \le 1$, this requires $\sum_{i=2}^{n}(\frac{g^{(i)}}{\lambda_i})^2 \le 1$. Otherwise $\lambda \ne 0$ and $\|s\| = 1$, and $\lambda$ is the solution of $\sum_{i=2}^{n}(\frac{g^{(i)}}{\lambda_i})^2 = 1$, such $\lambda > 0$ must exists if $\sum_{i=2}^{n}(\frac{g^{(i)}}{\lambda_i})^2 > 1$.

iv) If $\lambda_1 < 0$ and $g^{(1)} \ne 0$, then $\lambda \ge -\lambda_1$ and $\|s\| = 1$. Because $g^{(1)} \ne 0$, which implies $\lambda_1 > -\lambda_1$. Immediately we know $s = -(H + \lambda \mathrm{I}_d)^{-1}g$ and $\lambda$ is the solution of $\sum_{i=1}^{n}(\frac{g^{(i)}}{\lambda_i + \lambda})^2 = 1$,

v) If $\lambda_1 < 0$ and $g^{(1)} = 0$. In this condition, $\lambda > 0$ and $\|s\| = 1$. By gradient condition, we see $(\lambda_1 + \lambda)s^{(1)} = 0$ so either $\lambda = -\lambda_1$ or $s^{(1)} = 0$ (or both). This is determined by $\|s\| = s^{(1)2} + \sum_{i=2}^{n}(\frac{g^{(i)}}{\lambda + \lambda_i})^2 = 1$, if $\sum_{i=2}^{n}(\frac{g^{(i)}}{\lambda_i - \lambda_1})^2 \le 1$ then it is appropriate to set $\lambda = -\lambda_1$ and $s^{(1)} = \pm\sqrt{1 - \sum_{i=2}^{n}s^{(i)2}}$. Otherwise if $\sum_{i=2}^{n}(\frac{g^{(i)}}{\lambda_i - \lambda_1})^2 > 1$ then it must be that $s^{(1)} = 0$ and $\lambda > -\lambda_1$ such that $\sum_{i=2}^{n}(\frac{g^{(i)}}{\lambda + \lambda_i})^2 = 1$ holds, one can see such equation has only one solution.

$\square$

### A.2  PROOF OF LEMMA 2

*Proof.* Suppose $s$ is a stationary point, by formulating the Lagrange multiplier and using KKT condition, there exists a $\lambda \ge 0$ such that $(H + \lambda \mathrm{I}_d)s + g = 0$, and further by complementary slackness if $\|s\| < 1$ then $\nabla f(s) = Hs + g = 0$ this leads to $\lambda = 0$, otherwise $\|s\| = 1$ so in both cases the first and second conditions in (4) are reached. According to Lemma 1 if $s$ is not global minimum then the third condition should be violated, implying $\lambda_1 + \lambda < 0$, by gradient condition $(\lambda_1 + \lambda)s^{(1)} + g^{(1)} = 0$, because $g^{(1)} \ne 0$ indicating $s^{(1)} \ne 0$, multiply both sides by $s^{(1)}$ we get $s^{(1)}g^{(1)} > 0$, that is the case for stationary points excluding global minimum. On the contrary for global minimum, we must have $s^{(1)}g^{(1)} \le 0$. $\square$

### A.3  PROOF OF THEOREM 3

*Proof.* Notice the projection onto sphere will not change the sign of $z_{t+1}^{(i)}$, so:

$$
\mathrm{sgn}\left(z_{t+1}^{(i)}g^{(i)}\right) = \mathrm{sgn}\left((1 - \eta\lambda_i)z_t^{(i)}g^{(i)} - \eta g^{(i)2}\right)
$$

$\eta_t < 1/\lambda_n$ ensures $1 - \eta\lambda_i > 0$ for all $i \in [n]$. From Assumption 2 we know $z_0^{(i)}g^{(i)} = -\alpha\frac{g^{(i)2}}{\|g\|} \le 0$, so $z_t^{(i)}g^{(i)} \le 0$ for all $t$. $\square$

### A.4  PROOF OF LEMMA 4

*Proof.* Define $w_t^{(i)} = z_t^{(i)}/(-\eta g^{(i)})$, then by iteration rule:

$$
w_{t+1}^{(i)} = (1 - \eta\lambda_i)w_t^{(i)} + 1,
\tag{13}
$$

solving this geometric series, we get:

$$w_t^{(i)} = (1 - \eta\lambda_i)^t \big(w_0^{(i)} - \frac{1}{\eta\lambda_i}\big) + \frac{1}{\eta\lambda_i} \tag{14}$$

suppose at $t$-th iteration, $w_t^{(i)} \geq w_{t+1}^{(i)}$ which is equivalent to:

$$w_0^{(i)} - \frac{1}{\eta\lambda_i} \geq (1 - \eta\lambda_i)\big(w_0^{(i)} - \frac{1}{\eta\lambda_i}\big) \tag{15}$$

because from Assumption 2 we know $w_0^{(i)} \geq 0$, if $w_0^{(i)} - \frac{1}{\eta\lambda_i} \leq 0$, i.e. $0 < \eta\lambda_i \leq 1/w_0^{(i)}$ then by (15) we know $1 - \eta\lambda_i \geq 1 \Leftrightarrow \eta\lambda_i \leq 0$ leading to a contradiction, so $w_0^{(i)} - \frac{1}{\eta\lambda_i} > 0$ and $1 - \eta\lambda_i \leq 1$. On the other hand, $\lambda_j \geq \lambda_i$ for $j \geq i$, so $1 - \eta\lambda_j \leq 1$, together with $w_0^{(j)} - \frac{1}{\eta\lambda_j} = z_0^{(i)} - \frac{1}{\eta\lambda_j} \geq w_0^{(i)} - \frac{1}{\eta\lambda_i} > 0$ we conclude:

$$w_0^{(j)} - \frac{1}{\eta\lambda_j} \geq (1 - \eta\lambda_j)\big(w_0^{(j)} - \frac{1}{\eta\lambda_j}\big) \Leftrightarrow w_t^{(j)} \geq w_{t+1}^{(j)} \text{ for } j \geq i.$$

For any $t$, suppose $i^* \in [n]$ is the smallest coordinate index such that $w_t^{(i^*)} \geq w_{t+1}^{(i^*)}$, which implies $w_t^i < w_{t+1}^i$ for any $i < i^*$ and $w_t^i \geq w_{t+1}^i$ for any $i \geq i^*$ (such a $i^*$ may not exist, but it doesn't matter). By analyzing the sign of $z_t$ we know:

$$\text{sgn}\left(z_t^i\big(z_t^{(i)} - z_{t+1}^{(i)}\big)\right) = \text{sgn}\left(w_t^{(i)}\big(w_t^{(i)} - w_{t+1}^{(i)}\big)\right) = \text{sgn}\left(w_t^{(i)} - w_{t+1}^{(i)}\right),$$

finally we have:

$$\begin{aligned}
z_t^\mathsf{T} A\nabla f(z_t) &= \frac{1}{\eta} \sum_{i=1}^{i^*-1} \lambda_i z_t^{(i)}(z_t^{(i)} - z_{t+1}^{(i)}) + \frac{1}{\eta} \sum_{i=i^*}^{n} \lambda_i z_t^{(i)}(z_t^{(i)} - z_{t+1}^{(i)}) \\
&\geq \frac{\lambda_{i^*-1}}{\eta} \sum_{i=1}^{i^*-1} z_t^{(i)}(z_t^{(i)} - z_{t+1}^{(i)}) + \frac{\lambda_{i^*}}{\eta} \sum_{i=i^*}^{n} z_t^{(i)}(z_t^{(i)} - z_{t+1}^{(i)}) \\
&\geq \frac{\lambda_{i^*}}{\eta} \sum_{i=1}^{n} z_t^{(i)}(z_t^{(i)} - z_{t+1}^{(i)}) \\
&\geq \beta z_t^\mathsf{T} \nabla f(z_t).
\end{aligned} \tag{16}$$

$\square$

## A.5 PROOF OF THEOREM 5

*Proof.* First of all, notice $\|\tilde{z}_{t+1}\|^2 = \|\tilde{z}_t\|^2 - 2\eta\tilde{z}_t^\mathsf{T}\nabla f(\tilde{z}_t) + \eta^2\|\nabla f(\tilde{z}_t)\|^2$, so it remains to show $\tilde{z}_t^\mathsf{T}\nabla f(\tilde{z}_t) \leq 0$. We prove this by induction rule, suppose $\tilde{z}_{t-1}^\mathsf{T}\nabla f(\tilde{z}_{t-1}) \leq 0$ and from $\tilde{z}_t = \tilde{z}_{t-1} - \eta\nabla f(\tilde{z}_{t-1})$ we know:

$$\tilde{z}_t^\mathsf{T}\nabla f(\tilde{z}_t) = \tilde{z}_{t-1}^\mathsf{T}\nabla f(\tilde{z}_{t-1}) - \eta\|\nabla f(\tilde{z}_{t-1})\|^2 - \underbrace{\eta\tilde{z}_{t-1}^\mathsf{T} A\nabla f(\tilde{z}_{t-1})}_{(1)}$$

$$+ \underbrace{\eta^2\nabla f(\tilde{z}_{t-1})^\mathsf{T} A\nabla f(\tilde{z}_{t-1})}_{(2)}.$$

From Lemma 4 we know $(1) \geq \beta\tilde{z}_{t-1}\nabla f(\tilde{z}_{t-1})$ and recall $\beta$ is the operator norm of $A$, we have $(2) \leq \beta\|\nabla f(\tilde{z}_{t-1})\|^2$, so:

$$\tilde{z}_t^\mathsf{T}\nabla f(\tilde{z}_t) \leq (1 - \beta\eta)\tilde{z}_{t-1}^\mathsf{T}\nabla f(\tilde{z}_{t-1}) - \eta(1 - \eta\beta)\|\nabla f(\tilde{z}_t)\|^2, \tag{17}$$

naturally, by choosing $\eta < 1/\beta$ we have $\tilde{z}_t^\mathsf{T}\nabla f(\tilde{z}_t) \leq 0$ for all $t$, based on this observation $\|\tilde{z}_{t+1}\|$ is monotone increasing. $\square$

## A.6 PROOF OF LEMMA 6

*Proof.* This directly follows from:

$$1 \leq \eta^2 g^{(1)2} w_{t^*+1}^{(1)2} = \tilde{z}_{t^*+1}^{(1)2} \leq \|\tilde{z}_{t^*+1}\|^2,$$

together with (14) immediately comes to (5). □

## A.7 PROOF OF THEOREM 7,8

See Theorem 4.3.1 and Theorem 4.5.6 in (Absil et al., 2009).

## A.8 CALCULATING $\lambda_{H,\min}$ AND $\lambda_{H,\max}$

This is mainly brute force calculation. By definition (7), we know for $\xi \in T_x\mathcal{M}$,

$$\langle \text{Hess } f(x)[\xi], \xi \rangle = \langle \text{Hess } (f \circ R_x)(0_x)[\xi], \xi \rangle = \frac{\mathrm{d}^2}{\mathrm{d}t^2} f(R_x(t\xi))\Big|_{t=0}, \tag{18}$$

we then expand $f(R_x(t\xi))$ to,

$$f(R_x(t\xi)) = \frac{\xi^\intercal H\xi \cdot t^2 + \xi^\intercal Hx \cdot t + x^\intercal Hx}{2\|x+t\xi\|_2^2} + \frac{g^\intercal x + g^\intercal \xi \cdot t}{\|x+t\xi\|_2^2}. \tag{19}$$

By differentiating $t$ twice and set $t=0$ (this can be done by software), we finally get

$$\langle \text{Hess } f(x)[\xi], \xi \rangle = -x^\intercal Hx + \xi^\intercal H\xi - g^\intercal x. \tag{20}$$

Thus the eigenvalue of Riemannian Hessian comes from definition:

$$\lambda_{H,\min} = \min_{\|\xi\|=1, \xi \in T_x\mathcal{M}} \langle \text{Hess } f(x)[\xi], \xi \rangle,$$

$$\lambda_{H,\max} = \max_{\|\xi\|=1, \xi \in T_x\mathcal{M}} \langle \text{Hess } f(x)[\xi], \xi \rangle.$$

# B SUPPLEMENTARY EXPERIMENTS

## B.1 MOBILENETS+CIFAR10

We apply MobileNets (Howard et al., 2017) to classify CIFAR10 data, MobileNets is a light-weight network that designed for mobile and embedded vision applications. Here we still use $\eta = 0.1$ as the step size for SGD, for TR we use step size $\eta' = 0.01$, the batch size is $B = 2048$. Other settings are the same with VGG16 experiment in the main text. We find the overall experiment outcome are quit similar to VGG16 network (see Figure 6).

| | Stage I | | Stage II | | Stage III | |
|---|---|---|---|---|---|---|
| | method | accuracy | method | accuracy | method | accuracy |
| Track 1 | SGD(50) | 74.57% | TR(100) | 80.22% | SGD(50) | 81.46% |
| Track 2 | TR(100) | 78.42% | SGD(50) | 79.69% | TR(50) | 79.06% |

Table 2: Testing accuracy in different stages. In Track 1 we first run 50 epochs SGD then 100 epochs TR followed by 50 epochs SGD; In Track 2 we change the order to first 100 epochs TR then 50 epochs SGD and end by 50 epochs TR, so both tracks run 200 epochs in total. Testing accuracy are reported by the end of each stage. Here we choose batch size $B = 2048$.

## B.2 VGG16+STL10

Now we change our dataset to STL10 but still using VGG16 model. The parameter setting are the same with VGG16+CIFAR10. We choose batch size $B = 1024$, note that STL10 has only 5000 training samples, so even the batch size is as small as 1k can have considerable generalization loss. As can be seen in Figure 7 and Table 3.

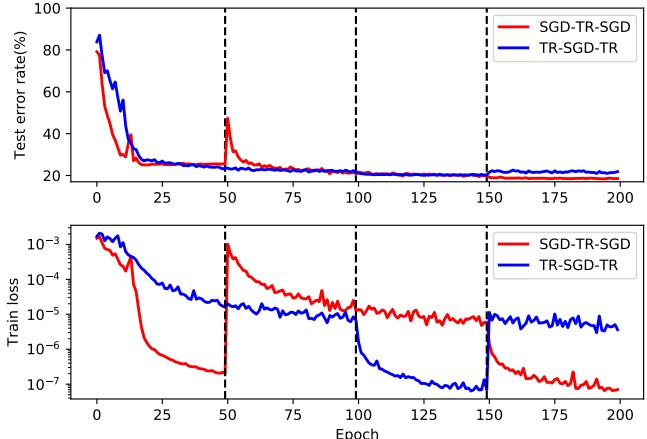

Figure 6: Experiments on two tracks as described in Table 2, we display training loss and test error w.r.t epochs, the batch size is $B = 2048$.

|         | Stage I |          | Stage II |          | Stage III |          |
|---------|---------|----------|----------|----------|-----------|----------|
|         | method  | accuracy | method   | accuracy | method    | accuracy |
| Track 1 | SGD(150) | 62.4%   | TR(300)  | 64.6%    | SGD(50)   | 66.0%    |
| Track 2 | TR(300)  | 63.6%   | SGD(50)  | 66.3%    | TR(150)   | 65.8%    |

Table 3: Testing accuracy in different stages. In Track 1 we first run 150 epochs SGD then 300 epochs TR followed by 50 epochs SGD; In Track 2 we change the order to first 300 epochs TR then 50 epochs SGD and end by 150 epochs TR, so both tracks run 500 epochs in total. Testing accuracy are reported by the end of each stage. Here we choose batch size $B = 1024$.

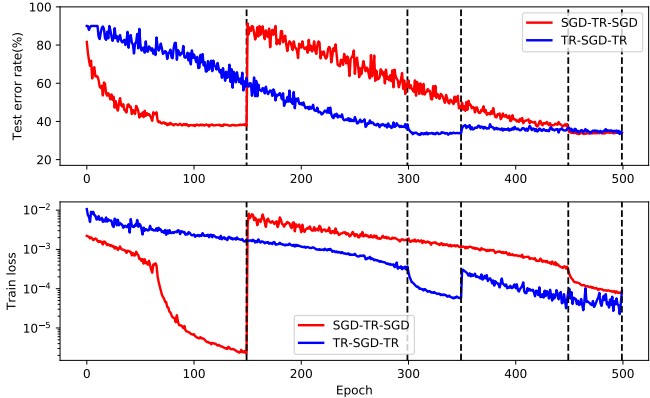

Figure 7: Experiments on two tracks as described in Table 3, we display training loss and test error w.r.t epochs, the batch size is $B = 1024$.

