# OpenReview forum: "Better Generalization by Efficient Trust Region Method"
_ICLR.cc/2018/Conference — Reject_

### Official Review · AnonReviewer1 · 2017-11-18
**I'm not fully convinced by the experiments, and the writing quality could be improved.**

**Rating:** 5
**Confidence:** 2

**Review:**

The paper proposes training neural networks using a trust region method, in which at each iteration a (non-convex) quadratic approximation of the objective function is found, and the minimizer of this quadratic within a fixed radius is chosen as the next iterate, with the radius of the trust region growing or shrinking at each iteration based on how closely the gains of the quadratic approximation matched those observed on the objective function. The authors claim that this approach is better at avoiding "narrow" local optima, and therefore will tend to generalize better than minibatched SGD. The main novelty seems to be algorithm 2, which finds the minimizer of the quadratic approximation within the trust region by performing GD iterations until the boundary is hit (if it is--it might not, if the quadratic is convex), and then Riemannian GD along the boundary.

The paper contains several grammatical mistakes, and in my opinion could explain things more clearly, particularly when arguing that the algorithm 2 will converge. I had particular difficulty accepting that the phase 1 GD iterates would never hit the boundary if the quadratic was strongly convex, although I accept that it is true due to the careful choice of step size and initialization (assumptions 1 and 2).

The central claim of the paper, that a trust region method will be better at avoiding narrow basins, seems plausible, since if the trust region is sufficiently large then it will simply pass straight over them. But if this is the case, wouldn't that imply that the quadratic approximation to the objective function is poor, and therefore that line 5 of algorithm 1 should shrink the trust region radius? Additionally, at some times the authors seem to indicate that the trust region method should be good at escaping from narrow basins (as opposed to avoiding them in the first place), see for example the left plot of figure 4. I don't see why this is true--the quadratic approximation would be likely to capture the narrow basin only.

This skepticism aside, the experiments in figure 2 do clearly show that, while the proposed approach doesn't converge nearly as quickly as SGD in terms of training loss, it does ultimately find a solution that generalizes better, as long as both SGD and TR use the same batch size (but I don't see why they should be using the same batch size). How does SGD with a batch size of 1 compare to TR with the batch sizes of 512 (CIFAR10) or 1024 (STL10)?

Section 4.3 (Figure 3) contain a very nice experiment that I think directly explores this issue, and seems to show that SGD with a batch size of 64 generalizes better than TR at any of the considered batch sizes (but not as well as the proposed TR+SGD hybrid). Furthermore, 64 was the smallest batch size considered, but SGD was performing monotonically better as the batch size decreased, so one would expect it to be still better for 32, 16, etc.

Smaller comments:

You say that you base the Hessian and gradient estimates on minibatched samples. I assume that the same is true for the evaluations of F on line 4 of Algorithm 1? Do these all use the same minibatch, at each iteration?

On the top of page 3: "M is the matrix size". Is this the number of elements, or the number of rows/columns?

Lemma 1: This looks correct to me, but are these the KKT conditions, which I understand to be first order optimality conditions (these are second order)? You cite Nocedal & Wright, but could you please provide a page number (or at least a chapter)?

On the top of page 5, "Line 10 of Algorithm 1": I think you mean Line 11 of Algorithm 2.

---

> ### Author Response · Authors · 2017-12-19
> **Response to AnonReviewer1**
>
> Thanks for your comments! We updated our submission according to your suggestions.
>
> We think you may have two major concerns: one is about how our trust region method avoids sharp minima, another is about the batch size. Here is our clarification:
>
> 1. “I had particular difficulty accepting that the phase 1 GD iterates would never hit the boundary if the quadratic was strongly convex”.
> ---
> Actually we claimed that “If the global minimum lies inside of the sphere then gradient descent itself is guaranteed to find it” (page 4, after the algorithm box). This means if the quadratic is strongly convex and the minimum lies in the interior of the ball, gradient descent in phase 1 will converge to it without hitting the boundary. This is because when we already know minimizer lies inside of the sphere, then the constraint ||s||<=1 is useless, so GD is equivalent to Prox-GD, and furthermore in theorem 5 (proven in appendix A5) we know the norm of GD iterate {||z_t||} is non-decreasing under our assumptions. If at some T, ||x_T||>=1 then ||x^*||<||x_T||, this contradicts to non-decreasing property. So we know it would never hit the boundary.
>
> 2. Honestly we don’t have theoretical explanations as to why stochastic TR method is better at generalization, since our claims are mostly based on empirical findings. However this is verified in all our experiments. To make it more clear, we plot the landscape of the function in Figure 4 in the revised version, and find that TR/Hybrid method converges to wide local minimums while SGD converges to a sharp local minimum. As an intuitive explanation, we think one reason might be the larger noise brought by subsampled Hessian.
>
> 3. “I don't see why they should be using the same batch size”
> ---
> Since the main point of our experiments is to compare the generalization ability across different algorithms, the effect of batch size should be controlled. Furthermore, under the same batch size, SGD and TR have similar computational costs per iteration and have the same level of parallelism. Also, this paper concerns especially about large batch case, since it is widely observed that SGD on large batch tends to find sharp minimum. We found our TR/hybrid method can mitigate this problem.
>
> 4. “How does SGD with a batch size of 1 compare to TR with the batch sizes of 512 (CIFAR10) or 1024 (STL10)”?
> ---
> Since deep networks rely heavily on batch normalization layer, it is not good to run batch size of 1. But we do have smaller batch result: for batch size=16 (which is nearly the minimum samples to make batch norm work), the accuracy lies between 83.3% on cifar-10(VGG), which is worse than SGD with batch size 64.
>
> 5. “Comparing accuracy on smaller batch size, like 32, 16, etc.”
> ---
> This is partly answered above that large batch case is more interesting and more actively researched. When we want to make use of many GPUs, one has to do large batch training. Even on smaller batch, the hybrid method still outperforms SGD, but the gap becomes small. In fact, when further decreasing the batch size, the test accuracy will drop, so the small batch size is not that interesting. For example, when batch size=16, accuracy of SGD is 83.3%, while Hybrid method is 83.4%. Typically the batch size is set to be 64~128 for one GPU, and larger (e.g., 1024 or larger) when training on multiple GPUs.
>
> As to your smaller comments:
> 1. “Evaluation of f(x_t)”.
> ---
> Please note that algorithm 1 is a very basic trust region method, we put it on to give readers a example of how our subproblem solver can be used. Actually, there are some stochastic version, for example:
> “Blanchet, J., Cartis, C., Menickelly, M. and Scheinberg, K., 2016. Convergence rate analysis of a stochastic trust region method for nonconvex optimization. arXiv preprint”
> Where they allow an estimation of function value, for example, calculated on minibatch. In our experiment, we use the same minibatch as computing gradient and Hessian.
> (In the updated version, we modify algorithm 1 to include this stochastic version)
>
> 2. M is the number of non-zero elements of matrix, or equivalently, complexity of matrix vector product.
>
> 3. Lemma 1 come from [Nocedal&Wright] Theorem 4.1, page 70 (in 2nd version).
>
> 4. Thanks for pointing out! We have fixed the typo in the updated version.

---

### Official Review · AnonReviewer2 · 2017-11-24
**A new stochastic method based on trust region (TR) is proposed. Experiments show improved generalization over mini-batch SGD. But the algorithm and its usefulness are neither developed nor explained well.**

**Rating:** 6
**Confidence:** 5

**Review:**

**I am happy to see some good responses from the authors to my questions. I am raising my score a bit higher.

Summary:
A new stochastic method based on trust region (TR) is proposed. Experiments show improved generalization over mini-batch SGD, which is the main positive aspect of this paper. The main algorithm has not been properly developed; there is too much focus on the convergence aspects of the inner iterations, for which there are many good algorithms already in the optimization literature. There are no good explanations for why the method yields better generalization. Overall, TR seems like an interesting idea, but it has neither been carefully expanded or investigated.

Let me state the main interesting results before going into criticisms:
1. TR method seems to generalize better than mini-batch SGD.
2. TR seems to lose generalization more gracefully than SGD when batch size is increased. [But note here that mini-batch SGD is not a closed chapter. With better ways of adjusting the noise level via step-size control (larger step sizes mean more noise) the loss of generalization associated with large mini-batch sizes can be brought down. See, for example: https://arxiv.org/pdf/1711.00489.pdf.]
3. Hybrid method is even better. This only means that more understanding is needed as to how TR can be combined with SGD.

Trust region methods are generally batch methods. Algorithm 1 is also stated from that thinking and it is a well-known optimization algorithm. The authors never mention mini-batch when Algorithm 1 is introduced. But the authors clearly have only the stochastic min-batch implementation of the algorithm in mind.

One has to wait till we go into the experiments section to read something like:
"Lastly, although in theory, we need full gradient and full Hessian to guarantee convergence, calculating them in each iteration is not practical, so we calculate both Hessian and gradient on subsampled data to replace the whole dataset"
for readers to realize that the authors are talking about a stochastic mini-batch method. This is a bad way of introducing the main method. This stochastic version obviously requires a step size; so it would have been proper to state the stochastic version of the algorithm instead of the batch algorithm in Algorithm 1.

Instead of saying that in passing why not explicitly state it in key places, including the abstract and title? I suggest TR be replaced by "Stochastic TR" everywhere. Also, what does "step size" mean in the TR method? I suggest that all these are fully clarified as parts of Algorithm 1 itself.

Trust region subproblem (TRS) has been analyzed and developed so much in the optimization literature. For example, the conjugate gradient-based method leading to the Steihaug-Toint point is so much used. [Note: Here, the gradient refers to the gradient of the quadratic model, and it uses only Hessian-vector products.] http://www.ii.uib.no/~trond/publications/papers/trust.pdf. The authors spend so much effort developing their own algorithm! Also, in actual implementation, they only use a crude version of the inner algorithm for reasons of efficiency.

The paper does not say anything about the convergence of the full algorithm. How good are the trust region updates based on q_t given the huge variability associated with the mini-batch operation? The authors should look at several existing papers on stochastic trust region and stochastic quasi-Newton methods, e.g., papers from Katya Scheinberg (Lehigh) and Richard Byrd (Colorado)'s groups.

The best-claimed method of the method, called "Hybrid method" is also mentioned only in passing, and that too in a scratchy fashion (see end of subsec 4.3):
"To enjoy the best of both worlds, we also introduce a “hybrid” method in the Figure 3, that is, first run TR method for several epochs to get coarse solution and then run SGD for a while until fully converge. Our rule of thumb is, when the training accuracy raises slowly, run SGD for 10 epochs (because it’s already close to minimum). We find this “hybrid” method is both fast and accurate, for both small batch and large batch."

Explanations of better generalization properties of TR over SGD are important. I feel this part is badly done in the paper. For example, there is this statement:
"We observe that our method (TR) converges to solutions with much better test error but
worse training error when batch size is larger than 128. We postulate this is because SGD is easy to overfit training data and “stick” to a solution that has a high loss in testing data, especially with the large batch case as the inherent noise cannot push the iterate out of loss valley while our TR method can."
Frankly, I am unable to decipher what is being said here.

There is an explanation indicating that switching from SGD to TR causes an uphill movement (which I presume, is due to the trust region radius r being large); but statements such as - this will lead to climbing over to a wide minimum etc. are too strong; no evidence is given for this.

There is a statement - "even if the exact local minima is reached, the subsampled Hessian may still have negative curvature" - again, there is no evidence.

Overall, the paper only has a few interesting observations, but there is no good and detailed experimental analysis that help explain these observations.

The writing of the paper needs a lot of improvement.

---

> ### Author Response · Authors · 2017-12-19
> **Response to AnonReviewer2**
>
> Thanks for your comments! We updated our submission according to your suggestions.
>
> [Ref1] Kohler, J.M. and Lucchi, A. Sub-sampled Cubic Regularization for Non-convex Optimization. ICML 2017.
> [Ref2] Blanchet, J., Cartis, C., Menickelly, M. and Scheinberg, K., 2016. Convergence rate analysis of a stochastic trust region method for nonconvex optimization. arXiv preprint
> [Ref3] Xu, P., Roosta-Khorasani, F. and Mahoney, M.W., 2017. Newton-type methods for non-convex optimization under inexact hessian information. arXiv preprint.
> [Ref4] Hazan, E. and Koren, T., 2016. A linear-time algorithm for trust region problems. Mathematical Programming, 158(1-2), pp.363-381.
>
> 1. First of all, let us explain the motivation of our work, this is not an article about the whole trust region method; like you said, its stochastic version with convergence rate is already developed in [Ref 1,2,3]. Independently, our innovation is mainly on developing the new solver for the trust region subproblem. For solving the trust region subproblem, several approximation methods such as Steihaug and Dogleg methods mentioned in section 1 and section 2.1 are used, but they can’t converge to the global minimum in nonconvex problems. More recently [Ref 4] proposed algorithms that converge to \epsilon-suboptimal solution in O(1/\sqrt{\epsilon}) time. We propose a new trust region subproblem solver (Algorithm 2), which converges faster than [Ref 4] in theory (linear convergence, O(log(1/\epsilon)) time), and we give rigorous theoretical proof for this subproblem solver.
>
> 2. “it would have been proper to state the stochastic version of the algorithm instead of the batch algorithm in Algorithm 1.”
> ---
> As to the organization of paper: The stochastic version of trust region method is very similar to the classical method, except that the stochastic one uses approximation of function value/gradient/Hessian. In the updated version, we elaborate more on stochastic version in Section 2.1 and mention in Algorithm 1 that g, H can be stochastic gradient and Hessian.
>
> 3. “The convergence of full-algorithm”.
> ---
> Since we do not specify the outer trust region method, the convergence property of the outer loop is guaranteed by standard analysis such as [Ref 1,2,3]. Our focus is on developing a better sub-problem solver, and apply the trust region method to solve deep neural networks. We have added the discussion of convergence rate in section 2.1.
>
> 4. “The best-claimed method of the method, called "Hybrid method" is also mentioned only in passing, and that too in a scratchy fashion”
> ---
> We agree that the hybrid method is important and should be formally introduced before experiments. We have added more details at the beginning of section 4.
>
> 5. “but statements such as - this will lead to climbing over to a wide minimum etc. are too strong; no evidence is given for this.”
> ---
> We notice that AnonReviewer 3 also has such concern, so we added another experiment in section 4.4 and Figure 4 to examine the wideness of local minimum, the result supports our claim that TR indeed helps to get a much wider minimum. We also calculate the distance of model parameters, we find solution of Hybrid method is very close to TR method, and both are far from SGD method. This supports our claim that Hybrid method is a refinement of TR method, and TR can escape the sharp local minimum.
>
> 6. “Frankly, I am unable to decipher what is being said here.”
> ---
> We have revised the sentence. We just want to explain “We observe that our method (TR) converges to solutions with much better test error when batch size is larger than 128. We postulate this is because SGD does not have enough noise to escape from a sharp local minimum, especially when large batch is used. “
>
> 7. “There is a statement - "even if the exact local minima is reached, the subsampled Hessian may still have negative curvature" - again, there is no evidence.”
> ---
> Yes, we agree that more evidence is needed to support this claim. Actually this is just our guess --- we think even in the local minimum where full Hessian is positive definite, the subsampled Hessian will not be positive since it is just a very rough approximation. We have withdrawn this claim. We also removed section 4.6 in our original version since it was just our intuition and guess but not very rigorous.

---

### Official Review · AnonReviewer3 · 2017-11-24
**See below.**

**Rating:** 6
**Confidence:** 3

**Review:**

The paper develops an efficient algorithm to solve the subproblem of the trust region method with an asymptotic linear convergence guarantee, and they demonstrate the performances of the trust region method incorporating their efficient solver in deep learning problems.  It shows better generation errors by trust region methods than SGD in different tasks, despite slower running time, and the authors speculate that trust-region method can escape sharp minima and converge to wide minima and they illustrated that through some hybrid experiment.
The paper is organized well.

1.  The result in Section 4.3 empirically showed that Trust Region Method could escape from sharp local minimum.  The results are interesting but not quite convincing.  The terms about sharp and wide minima are ambiguous.  At best, this provides a data point in an area that has received attention, but the lack of precision about sharp and wide makes it difficult to know what the more general conclusions are.  It might help to show the distance between the actual model parameters that those algorithms converge to.

2. As well know, VGG16 with well training strategy (learning rate decay) could achieve at least 92 percent accuracy. In the paper, the author only got around 83 percent accuracy with SGD and 85 percent accuracy with TR.  Why is this.

3. In section 4.2, it said "Although we can also define Hessian on ReLU function, it is not well supported on major platforms (Theano/PyTorch).  Likewise, we find max-pooling is also not supported by platforms to calculate higher order derivative, one way to walk around is to change all the max-pooling layers to avg- pooling, it hurts accuracy a little bit, albeit this is not our primary concern." It is my understanding that Pytorch support higher order derivative both for ReLu and Max-pooling.  Hence, it is not an explanation for not using ReLu and Max-pooling.  Please clarify

4. In section 4.3, the authors claimed that numerical diffentiation only hurts 1 percent error for second derivative. Please provide numerical support.

5. The setting of numerical experiments is not clear, e.g. value of N1 and N2.  This makes it hard to reproduce results.

5. It's not clear whether this is a theoretical paper or an empirical paper.  For example, there is a lot of math, but in Section 4.5 the authors seem to hedge and say "We give an intuitive explanation ... and leave the rigorous analysis to future works."  Please clarify.

---

> ### Author Response · Authors · 2017-12-19
> **Response to AnonReviewer3**
>
> Thanks for your comments! We updated our submission according to your suggestions.
>
> 1.  “The results are interesting but not quite convincing.  The terms about sharp and wide minima are ambiguous.  At best, this provides a data point in an area that has received attention, but the lack of precision about sharp and wide makes it difficult to know what the more general conclusions are.  It might help to show the distance between the actual model parameters that those algorithms converge to.”
> ---
> To show more evidence of the wide/sharp local minima, we added a similar experiment in section 4.4 (Figure 4) following [Keskar, 2016] that shows the loss and accuracy curve. We believe this could be served as the direct evidence of sharpness. Also, as you suggested, we calculated the distance between those models in Figure 4. As expected, the model computed by Hybrid  is very close to TR, and both of them are far away from the model computed by SGD. This indicates that Hybrid/TR models are quite different from the SGD model.
>
> 2. “In the paper, the author only got around 83 percent accuracy with SGD and 85 percent accuracy with TR.  Why is this.”
> ---
> For small batch (B=128), both of these methods get 87%-88% accuracy, ADAM has 86.6% accuracy. You said SGD has only 83% accuracy, so we think you were probably looking at large batch case.
> The loss of accuracy is because: 1) we didn’t do data augmentation, 2) we replace the max-pooling to avg-pooling and ReLU to our proposed SReLU. Both of them turn out to be worse than original layers. However, this accuracy is still decent compared with results in [Keskar 2016] (https://openreview.net/pdf?id=H1oyRlYgg) which uses standard VGG16 and has 89.24% accuracy.
>
> 3. “It is my understanding that Pytorch support higher order derivative both for ReLu and Max-pooling.  Hence, it is not an explanation for not using ReLu and Max-pooling.  Please clarify.”
> ---
> Although Pytorch support higher order derivative, we tested it by comparing the numerical computation of Hv with the auto-differentiation computed by Pytorch (0.2.0), and we find the results are inconsistent if the network contains ReLU and Max-Pooling.  To be safe, we changed those two layers to make sure both methods give the same value.
> The inconsistency might be due to numerical issues or some bugs in higher order auto-differentiation in Pytorch 0.2.0.
>
> 4. “In section 4.3, the authors claimed that numerical differentiation only hurts 1 percent error  for second derivative. Please provide numerical support.”
> ---
> We guess you mean “Experiments show that the relative error is controllable (~1%)”. This is measured by || Hv_analytic - Hv_numeric || / || Hv_analytic || < 0.01. For the fixed network, we calculate Hv product by both forward-backward and numerical differentiation, and then compute the average error (defined above) by randomly choosing many vectors v.
>
> 5. “The setting of numerical experiments is not clear, e.g. value of N1 and N2.  This makes it hard to reproduce results.”
> ---
> N1 and N2 are hyperparameters that depends on the problem scale. For the settings in our deep neural net experiments, the choice is discussed in the third paragraph of page 8, actually we do two inner iterations (N1=N2=1). We will release our code after reviewing process so the experiments will be reproducible.
>
> 6. “It's not clear whether this is a theoretical paper or an empirical paper.  For example, there is a lot of math, but in Section 4.5 the authors seem to hedge and say "We give an intuitive explanation ... and leave the rigorous analysis to future works."  Please clarify.”
> ---
> The first part of our paper proposes a new subproblem solver that is theoretically faster than [Hazan, 2014](https://arxiv.org/abs/1401.6757). Our method converges linearly while [Hazan 2014] converges sublinearly, and the proof for this part is rigorous (so it is quite theoretical for this part).
> The second part is mainly about empirical findings when we apply our method to deep neural networks, and we think this is as important as the first part. Since few papers about second order method evaluate their algorithms on deep networks, it is still unclear whether second order methods are useful to practitioners. However, currently we don’t have theoretical justification of why our method can avoid sharp local minima, so we only provide empirical evidence. Therefore for the second part we mentioned “We give an intuitive explanation of why trust region method avoids sharp local minima, and leave the rigorous analysis to future works.”
> But please note that in our updated version, we removed Sec. 4.5 since it doesn't have either theoretical or empirical guarantee.

---

### Decision · Program_Chairs · 2018-01-29
**ICLR 2018 Conference Acceptance Decision**

**Decision:**

Reject

**Comment:**

There are two parts to this paper (1) an efficient procedure for solving trust-region subproblems in second-order optimization of neural nets, and (2) evidence that the proposed trust region method leads to better generalization performance than SGD in the large-batch setting. In both cases, there are some promising leads here. But it feels like two separate papers here, and I'm not sure either individual contribution is well enough supported to merit publication in ICLR.

For (1), the contribution is novel and potentially useful, to the best of my knowledge. But as there's been a lot of work on trust region solvers and second-order optimization of neural nets more generally, claims about computational efficiency would require comparisons against existing methods. The focus on efficiency also doesn't seem to fit with the experiments section, where the proposed method optimizes less efficiently than SGD and is instead meant to provide a regularization benefit.

For (2), it's an interesting empirical finding that the method improves generalization, but the explanation for this is very hand-wavy. If second-order optimization in general turned out to help with sharp minima, this would be an interesting finding indeed, but it doesn't seem to be supported by other work in the area. The training curves in Table 1 are interesting, but don't really distinguish the claims of Section 4.5 from other possible hypotheses.